# Relationship between interoceptive sensibility and somatoform disorders in adults with autism spectrum traits. The mediating role of alexithymia and emotional dysregulation

**Elżbieta Zdankiewicz-Ścigała[1], Dawid Ścigała[2]\*, Joanna Sikora[2], Wanda Kwaterniak[1], Claudio Longobardi[3]**

1 Faculty of Psychology, SWPS University of Social Sciences and Humanities, Warsaw, Poland, 2 Institute of Psychology, The Maria Grzegorzewska University, Warsaw, Poland, 3 Università degli Studi di Torino, Turin, Italy

\* dscigala@aps.edu.pl

## Abstract

### Objective

The purpose of the study is to analyses the relationship between interoceptive sensibility and somatoform disorders among persons with Autism Spectrum Disorder (ASD). It has been assumed that the interoceptive sensibility is accompanied by a high level of alexithymia and emotion dysregulation in somatoform disorders.

### Methods

Persons under the care of the foundation helping people with ASD were asked to participate in the study. In total, 205 people took part in the research. The participants aged from 18 to 63 (M = 34.91; SD = 8.44). The ASD group comprised 79 persons (38.5% of subjects). The control group comprised 126 individuals (61.5% of subjects). Participants completed self-report questionnaires measuring autism (AQ), interoceptive sensibility (BPQ), alexithymia (TAS20), emotional dysregulation (DERS), and somatoform disorder (SDQ).

### Results

The analyses showed a moderation effect of the group, which indicates the existence of a relationship between interoceptive sensibility and somatoform disorders to the greater extent in the clinical group than in the control group. In addition, the serial multiple mediation model analysis allowed to verify the mediating effect of emotion dysregulation and alexithymia on the abovementioned relationship. The indirect effect, which assumed the mediating role of alexithymia turned out to be significant, contrary to the indirect effect where emotion dysregulation was a mediator in a situation where both variables were applied simultaneously.

**Data Availability Statement:** All relevant data are within the manuscript and its S1 File.

**Funding:** The author(s) received no specific funding for this work.

**Competing interests:** The authors have declared that no competing interests exist.

## Conclusions

Interoceptive sensibility correlated with level of alexithymia, in particular, difficulties in identifying and verbalizing emotions and emotion dysregulation in the lack of emotional awareness and lack of emotional clarity and is associated with somatoform disorders in the investigated group regardless of participants' belonging to the ASD or control group.

## Introduction

The purpose of the study is to verify dependencies between interoceptive sensibility and somatoform disorders in adults with Autism Spectrum Disorder (ASD). The main diagnostic feature of somatoform symptoms is the presence of physical symptoms, causes of which cannot be found in biological factors. Somatoform disorders often include sensory disorders or the loss of functions of specific organs, whose symptomatology implies the presence of somatoform or neurological disease which, in fact, cannot be diagnosed. Somatoform symptoms may occur both in the form of positive and negative symptoms. Negative somatoform symptoms refer to the impaired functioning of basic sensory functions or modalities, for example motion disorders, such as temporary paresis, paralysis, sensory disturbances, including vision, hearing, taste, smell, touch, pain, temperature or deep (proprioceptive) sensation. On the other hand, positive somatoform symptoms may be manifested as excessive activity, for example involuntary movements, intrusions in the form of taste, scent, feeling of body interaction or chronic pain, digestive system disorders, etc. The main criteria of somatoform disorders according with DSM– 5 are medically unexplained physical symptoms resulting in significant impairment to daily life [1]. Considering the etiology of these disorders, much importance is attached to the specific configuration of personality features. Numerous studies focus on the fact that somatoform symptoms presented by people may result from disorders in the cognitive processing of experienced body signals, affect and emotions [2–4].

### Interoception

Individuals, who have difficulty recognizing and verbalizing emotions, also experience problems with identifying and differentiating body conditions and functions such as heartbeat, breathing, body temperature, fatigue, hunger, thirst, satiety, muscle tension or pain [5]. Research demonstrates that the mechanism responsible for our emotional awareness significantly overlaps with the neural systems supporting our interoceptive awareness (IA) [6–11]. Craig [6, 7] suggested an broad approach to interoception as a system for perceiving somatic sensations such as heart rate, respiratory effort, fatigue, hunger, thirst, satiety, but also temperature, muscle tension, pain or itching (most frequently considered as exteroceptive information), as well as the representation of internal condition in the context of current activities [8]. Interoception refers to the ability to accurately perceive internal bodily processes, which comprise receiving, processing, and integrating body-relevant signals, together with external stimuli. Individual differences in interoception can be divided into three distinct dimensions: interoceptive accuracy (performance on objective tests of interoceptive accuracy), interoceptive sensibility (self-reported beliefs concerning one's own interoception) and interoceptive awareness (a metacognitive measure indexed by the correspondence between interoceptive accuracy and interoceptive sensibility) [10, 11]. The notion of interoceptive accuracy describes our capacity for identifying internal body sensations, whereas the notion of interoceptive sensibility applies

to our capacity for focusing on internal sensations and to take them into consideration, also from a cognitive point of view. In the light of present studies, interoception is associated with experiencing emotions, processing emotional stimuli, and activation of brain structures responsible for monitoring signals which come from the body, indicating the physiological and emotional condition [12]. There is increasingly more empirical evidence showing interoception as the basis of motivation, emotions, social cognition, and human self-awareness [5].

## Alexithymia and emotional dysregulation

Alexithymia, as found by many studies, is linked to non-secure attachment, especially emotional neglect in early childhood [13]. The exchange and social relationships in early life are essential for the emotional development of a child and later an adult. Schore [14] claims that especially important is the impact of attachment communication on the development of the right hemisphere involved in emotional processing, dealing with stress, self-regulation and shaping bodily-based implicit self. In the case of non-secure attachment, it is not only the disturbed psychological and emotional development, as has been pointed out so far, but also the inappropriate neurobiological development [14]. Experiences in a period critical for the development of a child's brain are encoded in rapidly developing right hemisphere (its frontal area in particular), affecting the ability to process negative emotions [15] or establishing a satisfactory relation between outside world and the self [16]. It is imperative to relate the child and caregiver to mutual responsiveness that takes the form of emotional tuning of the parent and child. Extremely important in this process are co called signified emotional responses, which mean that a parent adjusts to the child's emotional condition, expressing the same emotion but supplemented by signals which indicate that it is the reflection of the child's condition, and not the expression of the parent's pure emotions. The repetition of such situations enables the child to develop their own sense of Self, in particular, the awareness of own emotions. Those "signified" parent's emotional responses are fundamental to the development of the child's ability to accurately differentiate the physiological correlates of emotions, name emotions and regulate emotions. Experiencing a parent as unavailable and insensitive to the signals coming from the child for many years results in feeling chronic negative emotions, such as fear, anger or sadness. It is alexithymia that constitutes a persistent personality factor fostering deficits in the cognitive analysis of emotional arousal.

The main characteristics of alexithymia include (1) difficulty identifying feelings and distinguishing between feelings and bodily sensations of emotional arousal, (2) difficulty describing feelings to other people, (3) reduced capacity to fantasize and to imagine, (4) stimulus-bound, externally oriented cognitive style [17, 18], and, more recently, (5) low perspective-taking, as well as difficulty understanding and describing the emotions of others [19]. Emotion dysregulation includes difficulties in identifying and describing feelings, difficulties in distinguishing feelings from the bodily sensations of emotional arousal, impaired symbolization, as evidenced by the scarceness of fantasies and other imaginative activity, and a tendency to focus on external events rather than inner experiences (specific thought). According to Gratz and Roemer [20] emotional dysregulation is a multi-faceted construct involving: a lack of awareness, understanding, and acceptance of emotions; a lack of access to adaptive strategies for modulating the duration and/or intensity of aversive emotional experiences; an inability to control behaviors when experiencing emotional distress; and an unwillingness to experience emotional distress. It has also been conjectured that failure to experience complex emotional states is associated with exaggerated or dysregulated autonomic activation [21].

People who have difficulty processing their negative and positive emotions are more vulnerable to developing psychopathological symptoms [22]. One of the mechanisms that may be

responsible for the above is the asymmetry typical of people with high levels of alexithymia in experiencing negative over positive emotions. Negative emotions include anxiety, fear, contempt, disgust, anger, irritability, guilt or depression [22]. It is worth adding that individual differences in relation to negative vs. positive asymmetry are stable over time and trans-situational [22]. As a consequence, this entails problems with the regulation and self-regulation of emotions. As was noted above, an essential assumption underlying alexithymia theory is the fact that alexithymic individuals lack mental representations of emotions due to a deficit in the cognitive processing of these [23, 24]. The alexithymic individuals have limited capacities to use cognitive mechanisms to understand and regulate emotions lead them to focus on, amplify and misinterpret the bodily sensations accompanying emotional arousal. Porcelli and Taylor [25] have suggested that a failure to regulate and modulate stress-related emotions at the cognitive level may result in exaggerated physiological and behavioral responses to stressful situations and an increased vulnerability to disease [26].

Several studies indicate that one of the neuronal correlates of alexithymia is reduced neuronal activity in the emotional attention system, which includes the amygdala, fusiform gyrus and occipital cortex. Since the amygdala is involved in triggering emotional reactions and initiating ANS changes [25] the reduced activity of the amygdala in response to emotional stimuli will lead to a limited physiological response and disturbed somatosensory remapping. Persons who have difficulty identifying feelings may also have less access to information coming from the body, on which they could base spontaneous emotional reactions [26]. Kano et al. [27, 28] showed that persons with high levels of alexithymia present higher levels of anxiety and higher levels of adrenaline during visceral stimulation than people without alexithymia. The authors hypothesized that alexithymic individuals are more sensitive to unpleasant sensations from the body, and their higher autonomic reactivity is manifested in the increased activity of the right insula.

## Interoception, alexithymia and dysregulation emotion

In accordance with the adopted approach, factors, which foster somatization, include impairment interoception and alexithymia, understood as a disturbance in cognitive analysis and reinterpretation of stimuli both from the body and from the environment, as well as emotion dysregulation favoring the condition of continuous, chronic agitation of organism. An individual, who is unable to handle the excess of uninterpretable somatic conditions, high level of affective arousal, may cope using maladaptive defense mechanisms. Alexithymia may compound maladaptive behaviors. Such role of alexithymia has long been emphasized by Porcelli and Taylor [25] and by Lane [29]. A recent meta-analysis by De Gucht and Heiser [30] has demonstrated a small to moderate association between alexithymia and different self-report measures of somatization (e.g., health complaint scales; reflect = 0.23). The existing studies examining alexithymia in patients with somatoform disorders have yielded generally consistent evidence of increased levels of alexithymia in somatoform disorders.

Brewer et al. [31] investigated the association between alexithymia and self-reported non-affective interoceptive ability, as well as the extent to which people perceive a similarity between affective and non-affective states, in both control individuals and individuals reporting a diagnosis of a psychiatric condition. The findings showed that alexithymia was related to poor non-affective interoception and increased perceived similarity between affective and non-affective states, in both normal and clinical populations. Those with alexithymia not only have difficulty recognizing their own emotions (affective interoception) but also confuse these states with non-affective interoceptive states, such as hunger, tiredness, and arousal.

Numerous studies [32–35] have supported the coexistence of impairment interoception and alexithymia in the context of somatosensory amplification, i.e. exaggerated sensibility and excessive concentration on somatic sensations, and their misinterpretation [32] Messina et al. [33] indicate that the reception of body signals in alexithymia may be related to somatosensory amplification (SA). SA has been characterized as: 1) excessive attention and hyper-vigilance to somatic symptoms; 2) exaggerated sensitiveness to physical sensations; 3) misinterpretation of physical sensations interpreted as a sign of disease. Somatosensory amplification and somatization are generally associated with difficulty identifying and describing feelings (but less so with externally oriented thinking style) [33].

Barsky [34] defines this phenomenon as a tendency to feel the body tense, and its condition distressing, as well as an individual's tendency to focus on unpleasant somatic sensations, assign them a more pathological meaning than it actually is, and interpret them as the evidence of disease conditions. Such mechanism may operate on a vicious circle basis, thereby reinforcing deficits typical for alexithymia, i.e. difficulty in identifying and naming the sensations, leading additionally to the condition of chronic ANS arousal, impairment interoception. Dragoş and Tănăsescu [35] emphasize that the phenomenon of somatosensory amplification occurring in alexithymic individuals is strongly related to the mechanism of somatization, which, in their opinion, may be perceived as a cognitive disorder resulting from excessive perception of body sensations and their cognitive interpretation as a sign of disorder. The individual perceives many situations as threatening, which leads to chronic ANS arousal, hormonal stimulation, and it may finally result in further health problems. Typical for this type of somatic disorders is high variability of symptoms that a suffering individual complains about. Ailments may last for years and originate from different systems: digestive, cardiovascular, genitourinary, as well as they frequently affect skin, muscles and joints. They may include such symptoms as accelerated heartbeat, excessive sweating, dry mouth, flushing, gastrointestinal disorders, difficulty breathing, and chest discomfort. Typical for many persons is also persistent psychogenic pain, i.e. pain which may persist for numerous months, but cannot be explained by any disease process within the body.

## Alexithymia, interoception and autism spectrum disorders

Autism Spectrum Disorders (ASD) is a neurodevelopmental disorder characterized by difficulties with social communication and interaction, and restricted or repetitive patterns of behavior or interests [1]. Following the publication of the DSM-5 [1], a view of and approach to autism was developed positing it on a spectrum, no longer seen as a categorical condition. The term 'spectrum' is used for the reason of the heterogeneity in the presentation and severity of ASD symptoms, as well as in the skills and level of functioning of individuals with ASD [36]. It is now clear that genetic factors play a predominant role in its etiology [37]. Persons with autism spectrum present difficulties in social communication, which, according to many researchers, is related to deficits in emotion processing. This particularly refers to recognizing emotions in others or a decreased level of empathy [38]). Hill, et al., [39] have also supported that individuals with autism spectrum disorders have significantly greater difficulties in identifying and naming emotions than persons from general population.

A study by Leonardi et al., [40] found that parents of children with ASD show higher level of alexithymia, as measured with Toronto Structured Interview for Alexithymia, than children of neurotypical parents. Alexithymia, disrupting the appropriate recognition and understanding of emotions, including those expressed by facial expression, results in the misuse of emotions as important sources informing about the mental state and relationships with other people, and thus contributes to an increase in the number of stressful and stress-inducing

situations. A significant amount of data indicates a relationship between alexithymia and inter-oception [31] also in relation to persons with autism [41]. At the anatomical level, alexithymia and interoception are associated with impairment functioning of the same brain areas: insular cortex and cingulate cortex–the processing of emotions and the processing of interoceptive signals take place within the same neural system [31].

Researchers suggest that the impairment of interoception processes in ASD is caused by abnormalities in the way current experience updates predictions for the future [42]. Garfinkel et al. [11] demonstrated that individuals with ASD have reduced interoceptive accuracy (quantified using heartbeat detection tests) and exaggerated interoceptive sensibility (subjective sensibility to internal sensations on self-report questionnaires), reflecting an impaired ability to objectively detect bodily signals alongside an over-inflated subjective perception of bodily sensations. The divergence of these two interoceptive axes can be computed as a trait prediction error. This error correlates with deficits in emotion sensibility and occurrence of anxiety symptoms. They believe that these results indicate an origin of emotion deficits and affective symptoms in ASD. Atypical forms of interoceptive awareness have been noted in people who suffer from both alexithymia and ASD [43]. It has been advanced that alexithymia could be correlated with the atypical interoception which is noted in people suffering from ASD [36, 41, 44]; this might be reflected in their difficulties understanding their bodily states, as well as theirs and others' emotions [36, 45].

## Dissociation, alexithymia-regulation of emotions and somatization

Muskens et al. [46] conducted a meta-analysis to verify the prevalence of somatic disorders in people with attention deficit hyperactivity disorder (ADHD) and ASD. The main finding of this systematic review is that medical disorders in children with ASD and ADHD appear to be widespread, e.g., can manifest themselves across different medical areas, such as immunology, neurology, and gastroenterology. They suggest that these children require multidisciplinary medical services, including psychiatric help [46]. On the other hand, it has been proposed that certain features of ASD symptomatology may predispose this population to an increased risk of cumulative stressful life events and trauma exposure, and subsequent development of post-traumatic stress disorder (PTSD) [47–50]. Following exposure to trauma, PTSD may and does develop in certain individuals with ASD [50], with PTSD diagnosis found to be associated with suicidal thoughts and attempts within this population [51, 52]. Taylor and Gotham [50] suggest that contextual factors such as trauma might be important for the development of mood symptomatology in individuals with ASD. Mood and anxiety disorders are the most widespread psychiatric comorbidities in adolescents and adults with ASD [50].

Dissociation in the setting of chronic trauma is considered to be a coping strategy, at least initially. In the setting of childhood trauma, dissociation is thought to be a self-protective survival technique in which a child (or adult) slips into a dissociative state in order to escape fully experiencing trauma that is unbearable [53]. However, developmental mood and anxiety disorders, PTSD or physical symptoms strengthen psychological consequences of experienced traumas; in fact, they demonstrate symptoms that are distressing and confusing and that stand in the way of a fulfilling life. According to Nijenhuis [54], it is necessary to stress that the "psychological dissociation" and "somatoform dissociation" labels should not be taken to signify that only psychological dissociation is a mental phenomenon. Both labels relate to the ways in which dissociative symptoms may show themselves, not to their presumed cause. Somatoform dissociation designates dissociative symptoms that phenomenologically involve the body, and psychological dissociative symptoms are those that phenomenologically involve psychological variables [2]. The "somatoform" descriptor demonstrates that the physical symptoms are akin to, but cannot be explained by, a medical symptom or the direct effects of a substance. Thus

"somatoform dissociation" denotes phenomena that are manifestations of a lack of integration of somatoform functions, reactions, and experiences. Motor inhibitions and anesthesia/analgesia are somatoform dissociative symptoms that are akin to animal defensive reactions to major threat and injury [2, 53].

## Hypothesis and study rationale

It was hypothesized that the higher interoceptive sensibility in individuals with ASD is accompanied by a higher level of alexithymia on the emotions' identification and verbalization dimensions, as well as a higher dysregulation of emotions, especially in relation to deficits in realizing emotions and deficits in the clarity as to experienced emotions. However, to date, no studies have specifically assessed the relationships between interoceptive sensibility and somatoform disorders with the mediating role of alexithymia, and emotion dysregulation in adults with ASD (Fig 1).

To verify the above relationships, individuals with ASD have been selected due to the fact that disorders related to experiencing, understanding, and cognitive analysis of emotions in this group coincide with those observed in alexithymia. Mattila et al. [55] stated that alexithymia rates in the general population have been reported to be 9–17% for men and 5–10% for women whereas estimates are as high as 70% in some clinical groups [56]. Studies show that alexithymia has been found in nearly 50% of persons with ASD [57]. Gaig et al., [58] showed that alexithymia occurring both in the group with ASD and without this personality trait, is associated with the disturbance of appropriate differentiation of physiological arousal and thereby with distortions in the cognitive processing of experienced emotion and feelings. Therefore, it is alexithymia that is associated with the described disorders, not autism.

To the best of our knowledge, this is the first study aiming at: 1) testing whether somatoform disorders in the ASD patients might depend on interoceptive sensibility; 2) assessing the impact of alexithymia and dysregulation in the prediction of somatoform disorders using a battery of correlation studies, t-tests, moderation and mediation analyses. The serial multiple mediation model consistent with the theoretical findings and research results discussed above assumes an effect of alexithymia and dysregulation of emotions upon the relationship between interoceptive sensibility and somatoform disorders.

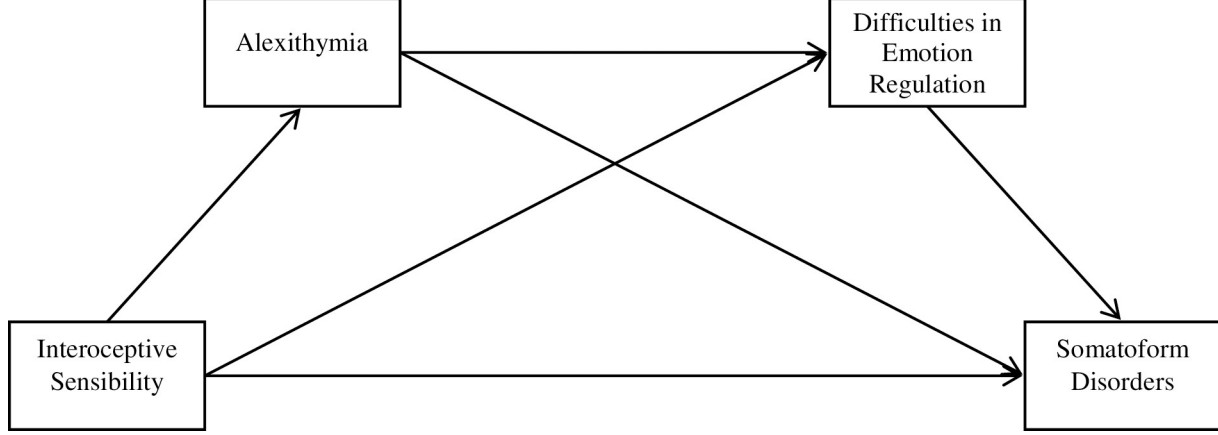

**Fig 1. A graphical model presenting the nature of dependencies between interoceptive sensibility (autonomic reactivity), alexithymia, and difficulties in emotion regulation in explaining an inclination to somatoform disorders.**

## Methods

### Participants

This study was conducted in accordance with the recommendations of the SWPS University of Social Sciences and Humanities Ethics Committee with written informed consent from all those who took part. All procedures during the study involving human participants were performed in accordance with the ethical standards of the institutional and/or national research committee and with the 1964 Helsinki declaration and its further amendments or comparable ethical standards. The project was approved by Institutional Review Board (IRB) of the SWPS University of Social Sciences and Humanities, Protocol No. 4/2019.

Persons under the care of the Foundation helping people with ASD located in Warsaw and other individuals not connected with the Foundation were asked to participate in the study, who constituted the control group. The study was carried out from April to August 2019. 205 people took part in the research including 157 women (76.6% of the total number of participants) and 48 men (23.4% of the total number of participants). The participants aged from 18 to 63 (M = 34.91; SD = 8.44). The clinical group included persons diagnosed with autism spectrum disorders, specifically Asperger's syndrome. The diagnosis has been additionally supported by the Autism-Spectrum Quotient. To be eligible for the clinical group, an individual must have been diagnosed with Asperger's syndrome and must have had at least 32 points in the AQ questionnaire. Ultimately, the clinical group comprised 79 persons, 72% women and 28% men, aged 20 to 45 (M = 35.12 SD = 6.32). The control group comprised individuals without any diagnosis of autism spectrum disorders and scoring less than 32 points in AQ questionnaire. The control group comprised 126 persons, 79% women and 21% men, aged 18 to 63 (M = 34.77 SD = 9.54).

### Measures

**The Toronto Alexithymia Scale–20 (TAS-20)** [59]. TAS-20 was applied to investigate the level of alexithymia. Other than the general level of alexithymia, the questionnaire allows us to estimate separate scales for aspects such as: "difficulties in verbalizing feelings" (DDF; "difficulties in identifying feelings" (DIF); "externally oriented thinking"(EOT). The questionnaire includes 20 test items. Each has a five-degree Likert scale (1-strongly disagree; 2-partially disagree; 3-no opinion; 4-partially agree; 5-strongly agree). The scale runs from 20 to 100 points. It is both a reliable and precise tool. In relation to the Polish version, Cronbach's α coefficient in non-clinical sample is 0.86 for the general score; 0.81 for the "difficulties in verbalizing feelings" scale; 0.75 for the "difficulties in identifying feelings" scale; and 0.64 for the "operational style of thinking" scale [60].

**The Autism-Spectrum Quotient (AQ)** [61]. AQ is the self-administered instrument for measuring a degree to which an adult with normal intelligence has the traits associated with the autistic spectrum. It comprises 50 questions, made up of 10 questions assessing 5 different areas: social skill; attention switching; attention to detail; communication; imagination. Each of the items listed above scores 1 point if the respondent records abnormal or autistic-like behavior either mildly or strongly. Abnormality = poor social skill, poor communication skill, poor imagination, exceptional attention to detail, poor attention-switching/strong focus of attention). Roughly 50% of the items were so worded to give a "disagree" response, and half an "agree" response, in a high scoring person with AS/HFA, to avoid a response bias either way. Also, items were randomized with respect to both the expected response from a high-scorer, and with respect to their domain. In the Polish version, Cronbach's α coefficient is 0.94 for the general score.

*Body Perception Questionnaire-Short Form (BPQ-SF)* [62] is a self-report measure of body awareness and autonomic reactivity. Was used to measure interoceptive sensibility. Its items are

based on the organization of the autonomic nervous system (ANS), a set of neural pathways connecting the brain and body. It comprises 46 questions. Item responses for both subscales are on a 5-point ordinal scale spanning never (1) to always (5). Body awareness was described by a single factor (sample items: "A swelling of my body or parts of my body", "Watering or tearing of my eyes", "Stomach and gut pains"). Autonomic reactivity reflected unique factors for organs above and below the diaphragm (sample items: "I have difficulty coordinating breathing and eating", "My heart often beats irregularly", "I have difficulty coordinating breathing with talking"). Psychometric properties were assessed from data in three samples: an American online study, a Spanish online study, and an American undergraduate student study (total n = 1320) [62]. In the Polish version, Cronbach's α coefficient is 0.91 for the general score; for body awareness Cronbach's α coefficient is 0.94; for ANS Cronbach's α coefficient is 0.91.

**Difficulties in Emotion Regulation Scale (DERS) [63].** The DERS items were chosen to reflect difficulties within the following dimensions of emotion regulation: (a) awareness and understanding of emotions, (b) acceptance of emotions, (c) ability to control impulsive behaviors and behave in accordance with desired goals when experiencing negative emotions, and (d) ability to use situationally appropriate emotion regulation strategies flexibly to modulate emotional responses as desired in order to meet individual goals and situational demands. The questionnaire comprises 36 test items. Each item has a five Likert scale (1- almost never; 5- almost always). The scale ranges from 36 to 180 points. The relative absence of any or all of these abilities would indicate the presence of difficulties in emotion regulation, or emotion dysregulation. The final dimension reflects an attempt to measure the flexible use of situationally appropriate strategies to modulate emotional responses. All of the DERS subscales (computed from the 6 factors obtained in the factor analysis) also had adequate internal consistency, with Cronbach's $\alpha > 0.80$ for each subscale. In the Polish version, Cronbach's α coefficient is 0.83 for the general score.

**Somatoform Dissociation Questionnaire (SDQ-20; [64]).** The SDQ-20 items were derived from a pool of 75 items describing clinically observed somatoform dissociative symptoms that in clinical settings had appeared upon reactivation of dissociative parts of the personality and that could not be medically explained. SDQ-20 is a self-report questionnaire which measures the severity of somatoform dissociation. Participants are asked about different physical symptoms or body experiences present in the past year. Participants are additionally asked if the experience is connected to any physical disease diagnosed by a physician and, if their answer is positive, to describe what that is. The items pertain to negative (e.g., analgesia) and positive dissociative phenomena (e.g., site-specific pain). The items are supplied with a Likert-type 5-point scale, ranging from "1 = this applies to me: not at all" to "5 = this applies to me: extremely". The total score is calculated by summing up the item scores and may range from 20 to 100. The internal consistency of the SDQ-20 is excellent ([64]: Cronbach's alpha 0.95). In the Polish version, Cronbach's α coefficient is 0.89 for the general score.

## Procedure

The study was carried out with each person individually. A question regarding the possibility of conducting a questionnaire survey was sent by email to the Foundations which help adults with ASD. Having obtained the consent, the researcher visited the Foundation's office, introduced themselves, and invited persons who were willing to participate in the study. At first, each participant received the form of informed consent to participate in the study. Then questionnaires to be filled in were provided to participants in envelopes. Questionnaires were completed in random order. After completing, the questionnaires were returned to the researcher. In order to complete the control group, an announcement was published on SWPS University

website with the invitation to participate. The willing persons contacted via a given email address. After making an appointment at the University Campus, as in the case of the clinical group, the respondents were given an informed consent form to participate in the study, and after that they were given an envelope containing the questionnaire. Upon completion, each participant was thanked for taking part in the study.

### Data analysis

A statistical analysis to test the posited hypotheses was undertaken using IBM SPSS Statistics, version 26. Key descriptive statistics were undertaken using the software, making it possible to study the distributions of successive measured variables. Parametric tests were performed on all variables as the skewness values did not exceed the conventional absolute value of 1. In the first place, a series of correlation analyses was performed to assess the relationship between somatoform disorders and all other variables. Further, a series of t-tests was performed to determine differences between the clinical and control groups in terms of alexithymia, emotion dysregulation, and interoceptive sensibility dimensions. The next step was to carry out a moderation analysis, where a (clinical/control) group was the moderator, using the A. F. Hayes PROCESS macro (model 1) [65] to determine to what extent the fact of being diagnosed with ASD affects the relationship between interoceptive sensibility (autonomic reactivity) and somatoform disorders. Subsequently, the mediating role of all dimensions of alexithymia and emotion dysregulation was verified separately using the A. F. Hayes PROCESS macro (model 4). Finally, a method was applied that combined the relationship of interoceptive sensibility (autonomic reactivity) and somatoform disorders with the mediating role of alexithymia and emotion dysregulation. Due to the high level of correlation between mediators, the serial multiple mediation model A. F. Hayes PROCESS macro (model 6) was performed since this method assumes a strong relation between mediators [65]. In addition, it allows to control the impact of a single mediator, as well as to determine the simultaneous impact of both mediating variables on the relation between an independent and dependent variable.

### Results

Descriptive statistics for the study variables and for differences between the groups are reported in Table 1. Before more advanced statistical analyses were commenced, analyses were undertaken with the aim of verifying if statistically significant differences exist between the group of ASD and the control group as far as interoceptive sensibility, alexithymia, dysregulation emotions, and somatoform disorders are concerned. Descriptive statistics, t-tests results and correlations are reported in Tables 1 and 2.

Correlation analyses which took into account the relationship between the examined variables in relation to somatoform disorders were carried out for the whole group of subjects due to the fact that in the initial analyses the correlation strength for individual clinical and control groups was checked separately, and the obtained values did not differ significantly. Respectively, for the clinical group, the moderate of the relationship between Autism scale and somatoform disorders (r = 0.522 p < 0.01); and for the control group (r = 0.457, p < 0.01). As it results from the data presented in Table 2, a moderate positive relationship was shown between the severity of autism spectrum disorders (ASD) and the symptoms resulting from somatoform disorders (r = 0.550 p < 0.01). Similarly, a moderate correlation was obtained for: interoceptive sensibility (autonomic reactivity) (r = 0.512, p < 0.01); difficulties in identifying emotions (r = 417, p <0.01); alexithymia (r = 0.386, p < 0.01); lack of emotional clarity (r = 0.370, p < 0.01), and difficulties in emotion regulation as a general result (r = 0, 360, p <0.01). The obtained results formed the basis for more advanced statistical analyses. As the

**Table 1. Descriptive statistics and t-tests of the clinical (ASD) and control group.**

| | Groups | | | | t-tests | | |
| | Clinical | | Control | | *t* | *p* | *Cohen`s d* |
| | **M** | **SD** | **M** | **SD** | | | |
|---|---|---|---|---|---|---|---|
| Age | 35.12 | 6.32 | 34.78 | 9.55 | *0.282* | *0.778* | *.04* |
| **AQ** | | | | | | | |
| Autism | 39.33 | 4.52 | 17.75 | 7.79 | *25.080* | *<**0.001*** | ***3.52*** |
| **BPQ** | | | | | | | |
| Body Awareness | 24.18 | 2.61 | 24.32 | 2.54 | *0.362* | *0.718* | *.05* |
| Autonomic reactivity | 15.51 | 3.77 | 10.94 | 5.15 | *7.311* | *<**0.001*** | ***1.04*** |
| **TAS** | | | | | | | |
| Difficulty Describing Feelings | 18.71 | 3.79 | 13.10 | 5.01 | *9.100* | *<**0.001*** | ***1.30*** |
| Difficulty Identifying Feeling | 26.58 | 5.44 | 19.93 | 6.71 | *7.776* | *<**0.001*** | ***1.13*** |
| Externally-Oriented Thinking | 18.04 | 5.19 | 17.07 | 4.25 | *1.389* | *0.148* | *.23* |
| Alexithymia—Total | 63.33 | 11.72 | 50.10 | 12.70 | *7.477* | *<**0.001*** | ***1.05*** |
| **SDQ** | | | | | | | |
| Somatoform Dissociation | 38.46 | 12.79 | 25.99 | 5.94 | *8.131* | *<**0.001*** | ***1.63*** |
| **DERS** | | | | | | | |
| Non-Acceptance of Emotional Responses | 16.95 | 6.50 | 14.03 | 6.01 | *3.277* | *<**0.001*** | ***.46*** |
| Difficulties Engaging in Goal-Directed Behavior | 18.94 | 4.64 | 14.81 | 5.04 | *5.879* | *<**0.001*** | ***.82*** |
| Impulse Control Difficulties | 16.46 | 5.98 | 14.25 | 5.36 | *2.736* | *<**0.05*** | ***.38*** |
| Lack of Emotional Awareness | 19.35 | 5.39 | 15.52 | 4.64 | *5.398* | *<**0.001*** | ***.76*** |
| Limited Access to Emotion Regulation Strategies | 25.99 | 6.56 | 20.19 | 7.91 | *5.444* | *<**0.001*** | ***.76*** |
| Lack of Emotional Clarity | 15.03 | 4.09 | 10.88 | 3.73 | *7.461* | *<**0.001*** | ***1.05*** |
| Difficulties in Emotion Regulation Scale—Total | 112.71 | 22.32 | 89.69 | 25.98 | *6.740* | *<**0.001*** | ***.99*** |

first one, a moderation analysis was carried out, which took into account the division of the clinical and control groups (see Figs 2 and 3).

**Table 2. The analysis of correlations between particular dimensions of interoceptive sensibility, alexithymia, difficulties in emotion regulation, autism and the intensification of somatoform disorders.**

| | Somatoform Disorders |
|---|---|
| Autism | *,550**[**]* |
| Interoceptive sensibility (Body Awareness) | *-,008* |
| Interoceptive sensibility (Autonomic reactivity) | *,512**[**]* |
| Difficulty Describing Feelings | *,382**[**]* |
| Difficulty Identifying Feeling | *,417**[**]* |
| Externally-Oriented Thinking | *,084* |
| Alexithymia—Total | *,386**[**]* |
| Non-Acceptance of Emotional Responses | *,234**[**]* |
| Difficulties Engaging in Goal-Directed Behavior | *,268**[**]* |
| Impulse Control Difficulties | *,295**[**]* |
| Lack of Emotional Awareness | *,251**[**]* |
| Limited Access to Emotion Regulation Strategies | *,279**[**]* |
| Lack of Emotional Clarity | *,370**[**]* |
| Difficulties in Emotion Regulation Scale—Total | *,360**[**]* |

[*]-p<0,05;

[**]-p<0,01.

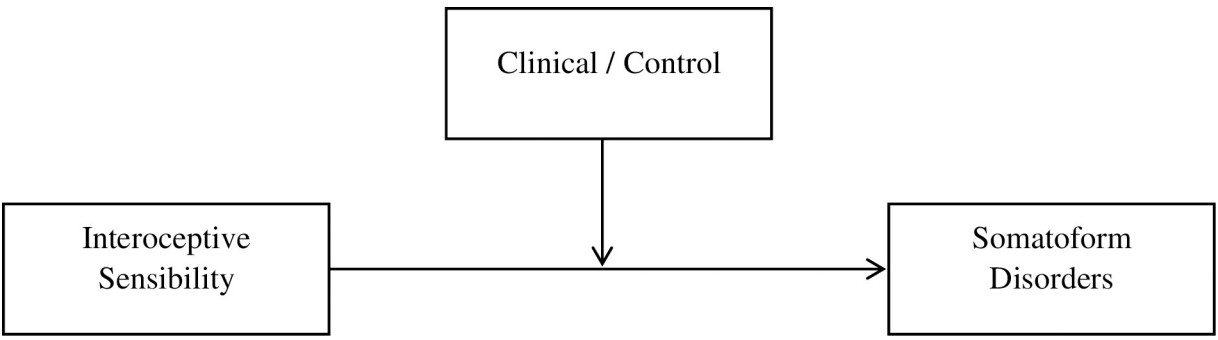

**Fig 2. A graphical model presenting the nature of dependencies in moderation model.**

The analysis demonstrated a significant interactive effect $F_{(1.201)} = 6.6892$; $p < 0.05$. Interestingly, for both study groups, the result is statistically significant, i.e. the group with ASD (standardized effect = 0.6037; 95% CI [.37; .84]), and the control group (standardized effect = 0.2474; 95% CI [.11; .38]). Noteworthy is the significantly higher effect strength in the clinical group and the relationship between the results in both groups (see Fig 3). In the case of the ASD group, the relationship between interoceptive sensibility (autonomic reactivity) and somatoform disorders was found as regards medium and elevated results on both scales. The higher the result on the interoceptive sensibility (autonomic reactivity) scale in the group of subjects with ASD, the higher the results on the scale examining somatoform disorders. In the case of the control group, the relationship between the interoceptive sensibility (autonomic reactivity) and somatoform disorders was demonstrated, but in relation to low scores on both scales. A different, yet also significant, distribution of dependencies between the studied

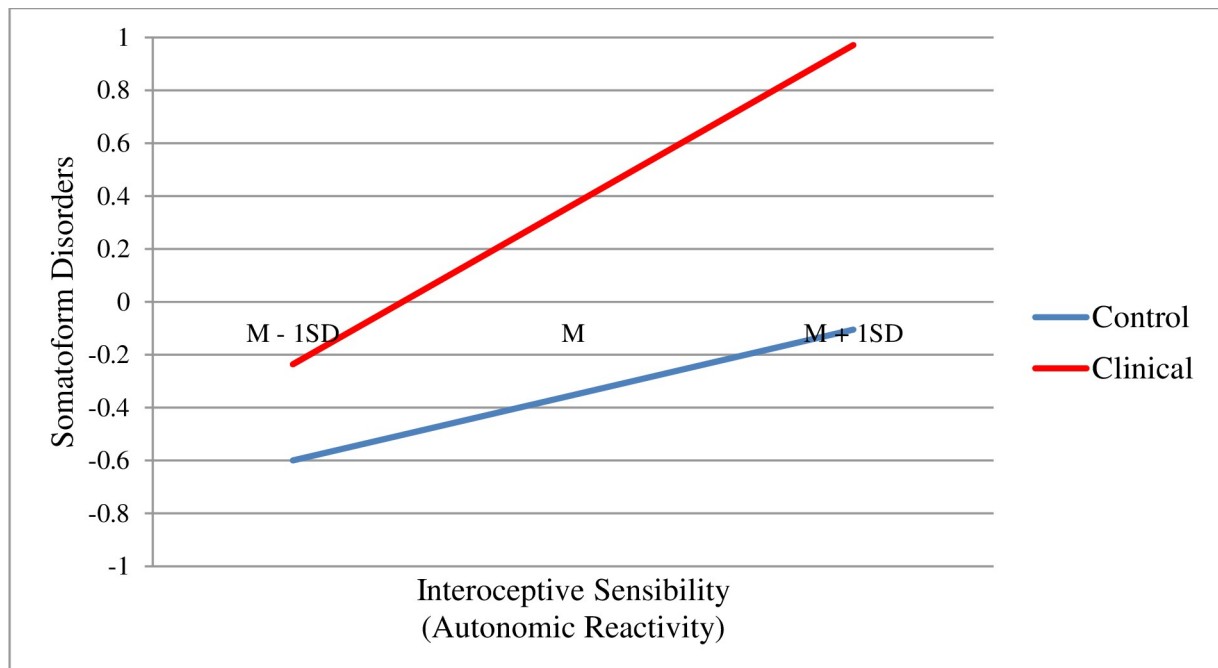

**Fig 3. The moderation effect of the clinical and control group on the relation between interoceptive sensibility (autonomic reactivity) and somatoform disorders.** Interoceptive sensibility (autonomic reactivity) was measured by Body Perception Questionnaire-Short Form (BPQ-SF). Somatoform disorders was measured by Somatoform Dissociation Questionnaire (SDQ-20).

variables in this group was demonstrated. The increased interoceptive sensibility (autonomic reactivity) in this group also results in somatoform disorders, however with a significantly lower intensity than in case of the clinical group.

The obtained results, which support the direct impact of interoceptive sensibility (autonomic reactivity) on somatoform disorders in the examined group, further formed the basis for the development of three mediation models. Due to the same direction of dependencies, demonstrated in the analysis of moderations, both in the control and clinical groups, the subsequent analyses of mediations were performed for the entire group without division.

The first to be analysed was the mediation effect of individual components of alexithymia on the relationship between interoceptive sensibility (autonomic reactivity) and somatoform disorders (see Fig 4). The model in which three mediators (alexithymia subscales) were introduced separately proved fit well with the data (F (4.200) = 22.9385; p < 0.001) and explains 31% of the variance in the level of somatic disorders resulting from somatoform disorders. Difficulties in verbalizing emotions (standardized effect = 0.775; 95% CI [.03; .15]), and difficulty in identifying emotions (standardized effect = 0.1063; 95% CI [.02; .19]) have been proven to be important mediators.

In the next step, an analysis of the mediation impact of difficulties in emotion dysregulation on the relationship between interoceptive sensibility and somatoform disorders was carried out (see Fig 5). The mediation model with six mediators (DERS subscales) suited the data (F (7,197) = 11.7974; p < 0.001) and explains 29% of the variance in the results of somatic disorders resulting from somatoform disorders. Two dimensions related to problems in regulating emotions were shown to be significant mediators, i.e. lack of emotional awareness (standardized effect = 0.0325; 95% CI [.004; .08]) and lack of emotional clarity (standardized effect = 0.0790; 95% CI [.008; .17]).

Based on the results obtained from both mediations which separately verified the effect of alexithymia and difficulties in regulating emotions in the relationship between interoceptive sensibility and somatoform disorders, another mediation was carried out. This time it was a Serial-Multiple Mediation model including alexithymia and difficulties in emotion

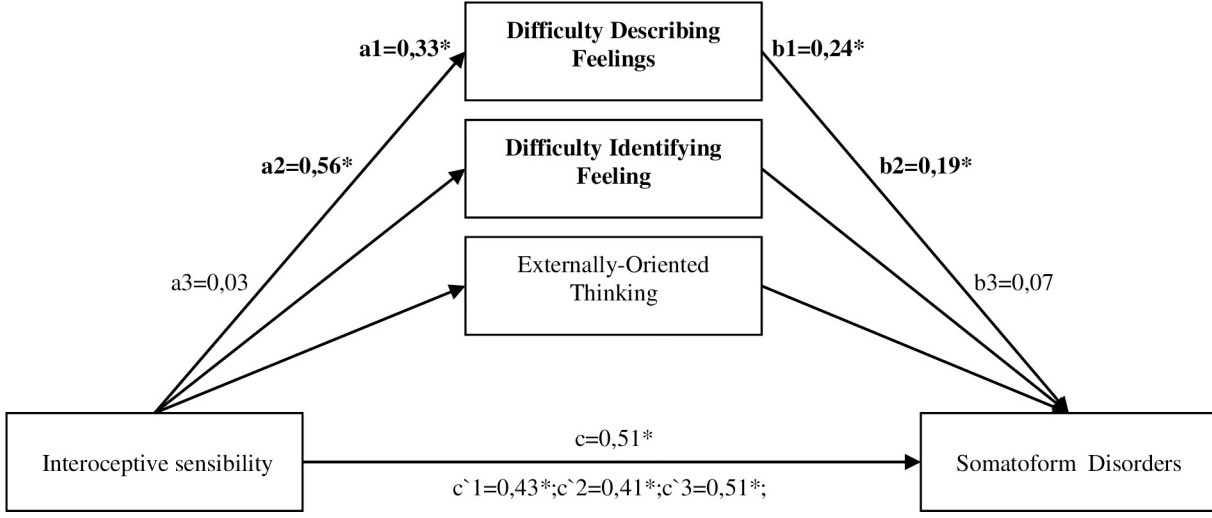

**Fig 4. The model of mediation for the tendency toward alexithymia between the impact of interoceptive sensibility (autonomic reactivity) on the level of somatoform disorders.** *-p<0,05; Significant mediation relation are marked in bold. Direct effect–X on Y without the influence of M1 –c. Indirect effect of X on Y through M—a1,b1; a2,b2; a3,b3. Direct effect–X on Y including the influence of M–c‘1;c‘2;c‘3. Interoceptive sensibility (autonomic reactivity) was measured by Body Perception Questionnaire-Short Form (BPQ-SF). Somatoform disorders was measured by Somatoform Dissociation Questionnaire (SDQ-20). Alexithymia was measured by The Toronto Alexithymia Scale– 20 (TAS-20).

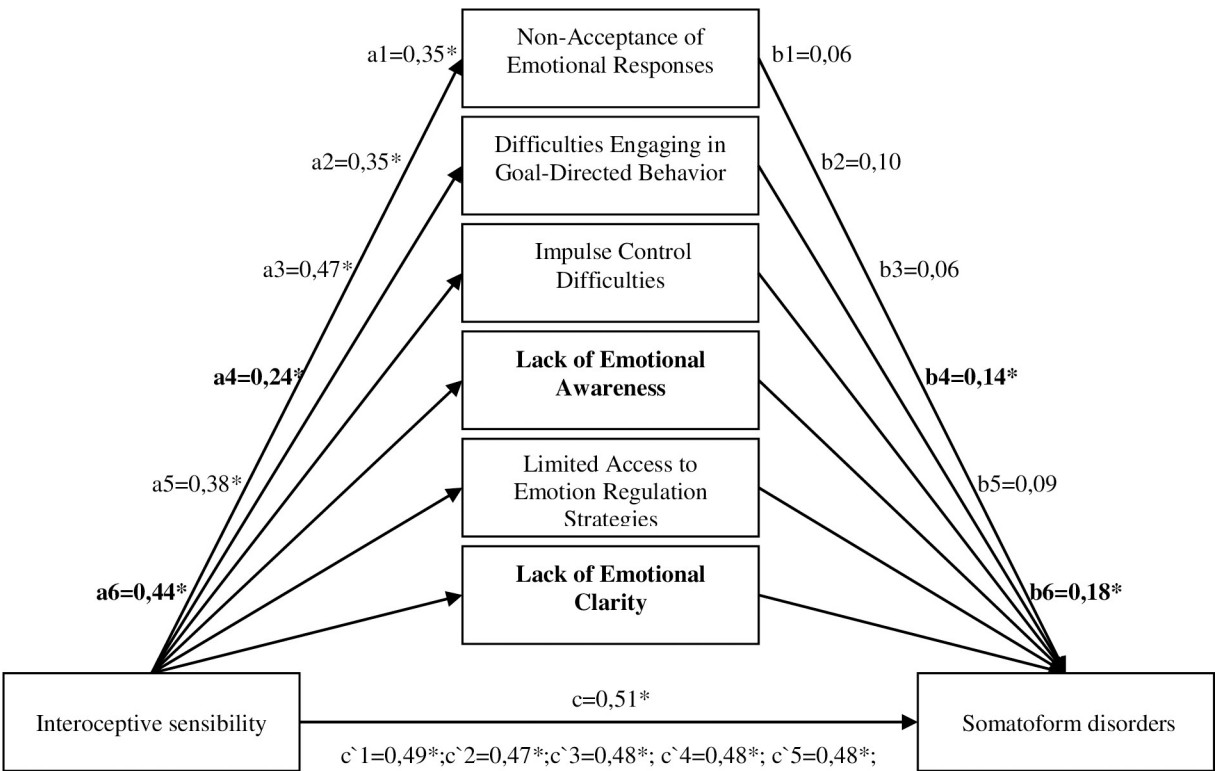

**Fig 5. The model of mediation for the tendency toward difficulty in regulation emotion between the impact of interoceptive sensibility (autonomic reactivity) on the level of somatoform disorders.** *-p<0,05; Significant mediation relation are marked in bold. Direct effect–X on Y without the influence of M1 –c. Indirect effect of X on Y through M—a1,b1; a2,b2; a3,b3; a4,b4; a5,b5; a6,b6. Direct effect–X on Y including the influence of M–c'1; c'2;c'3,c'4,c'5,c'6. Interoceptive sensibility (autonomic reactivity) was measured by Body Perception Questionnaire-Short Form (BPQ-SF). Somatoform disorders was measured by Somatoform Dissociation Questionnaire (SDQ-20). Difficulty in regulation emotion was measured by Difficulties in Emotion Regulation Scale (DERS).

dysregulation as mediators on the relation between interoceptive sensibility and somatoform disorders (see Fig 6).

The model is fitted well with the data $F_{(3.201)} = 28.4191$; p<0.001 and explains 29.8% variability of the dependent variable. As presented in Fig 6, the direct effect of the relationship between interoceptive sensibility (autonomic reactivity) and somatoform disorders is significant (c = .51. SE = .06, t = 8.50, p < .0001) (Step 1). The following step (step 2) is determining the relationship between interoceptive sensibility (autonomic reactivity) and alexithymia, which is also important (a1 = .42, SE = .06, t = 6.61, p < .0001). Another step (step 3) refers to the relationship of interoceptive sensibility (autonomic reactivity) with difficulties in emotion dysregulation (a2 = .22, SE = .05, t = 4.25, p < .0001) and alexithymia with difficulties in emotion dysregulation (d21 = .63, SE = . 05, t = 12.30, p < .0001). In the next step (step 4), the relationship between alexithymia and somatoform disorders was verified (b1 = .20, SE = .09, t = 2.32, p < .001), as well as the relationship between difficulties in emotion dysregulation and the dependent variable (b2 = .01, SE = .09, t = .11, p = .91). After introducing the independent variable and both mediators into the model simultaneously (step 5) the direct effect of interoceptive sensibility (autonomic reactivity) on somatoform disorders decreased, but it still statistically significant remained (c'= .42, SE = .07, t = 6.22, p < .0001).

The analysis of indirect effects based on 95% confidence intervals using 10,000 bootstrapping shows that the first indirect effect relating to the relationship between interoceptive

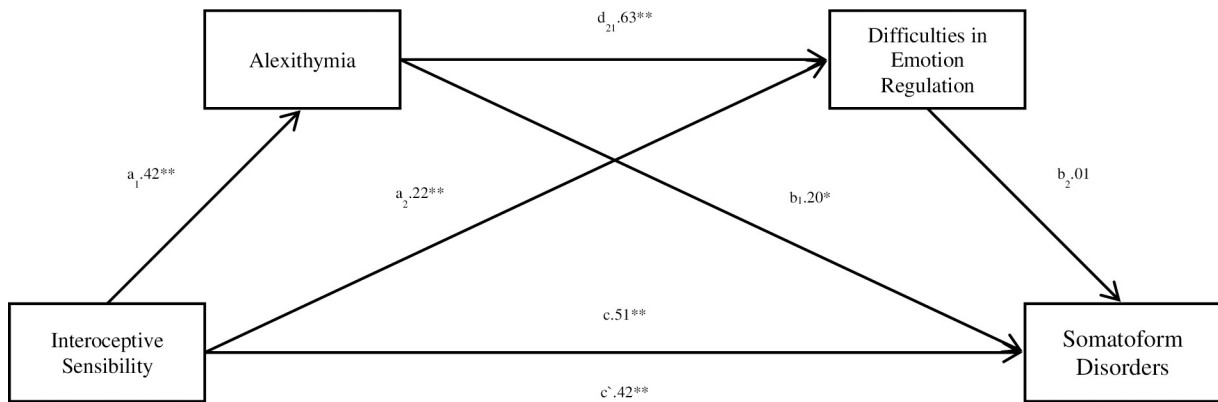

**Fig 6. Serial multiple mediational model for the relation between an interoceptive sensibility (autonomic reactivity) and somatoform disorders, where alexithymia and difficulties in emotion regulation constitute mediators.** (c) A direct effect of the impact of interoceptive sensibility on the somatoform disorders. (a1, b1) An indirect effect of the impact of interoceptive sensibility on the somatoform disorders, including alexithymia. (a2, b2) An indirect effect of the impact of interoceptive sensibility on the somatoform disorders, including Difficulties in Emotion Regulation. (a1, d21, b2) An indirect effect of the impact of interoceptive sensibility on the somatoform disorders, including Alexithymia and Difficulties in Emotion Regulation. (c') A direct effect of the impact of interoceptive sensibility on the somatoform disorders, taking account of the impact of both mediators. *-p<0,05; **-p<0,01. Interoceptive sensibility (autonomic reactivity) was measured by Body Perception Questionnaire-Short Form (BPQ-SF). Somatoform disorders was measured by Somatoform Dissociation Questionnaire (SDQ-20). Difficulty in regulation emotion was measured by Difficulties in Emotion Regulation Scale (DERS). Alexithymia was measured by The Toronto Alexithymia Scale– 20 (TAS-20).

sensibility (autonomic reactivity) and somatoform disorders with the mediating role of alexithymia (a1,b1) is significant (point estimate b = .08. 95% CI [.01; .16]) (Fig 6). In the case of the second indirect effect, where difficulties in emotion dysregulation is a mediator (a2, b2), the result turned out insignificant (point estimate b = .002. 95% CI[-.04; .05]) (Fig 6). The last indirect effect taking into account both alexithymia and difficulties in emotion dysregulation (a1, d21, b2) was also statistically insignificant (point estimate b = .003. 95% CI [-.04; .05]). The overall effect, which refers to the difference between the direct and indirect effect (c / c'), after taking both mediators into account, is significant (point estimate b = .09. 95% CI [0.01;0.18]).

To summarize, with Serial-Multiple Mediation model, a number of dependencies between interoceptive sensibility (autonomic reactivity) and somatoform disorders was demonstrated also taking into account the mediating role of alexithymia and emotion dysregulation, however the significant effects in this model were the direct and total effects, as well as the indirect dependency taking into account alexithymia as a mediator, and the effects taking into account emotion dysregulation and both mediators simultaneously are insignificant.

## Discussion

Our hypotheses, which state that alexithymia and emotion dysregulation play a mediating role between interoceptive sensibility and somatic disorders, have been supported both in the group of persons with ASD and in the control group. Interesting is the fact that the direction of relationship between the investigated variables is the same in both groups. However, in the case of persons with ASD, higher level of both interoceptive sensibility and alexithymia, as well as somatoform symptoms have been found. It was stated that the increased sensibility to interoceptive signals, combined with the difficulty in assigning those sensations to emotions causes them to be misinterpreted, reversely enhancing sensations, and, contributes to somatic symptoms. The increased interoceptive sensibility particularly relates to alexithymia, as was supported by the mediation analyses. Two factors of alexithymia are specifically important, i.e.:

difficulty in identifying emotions and difficulty in verbalizing emotions. The mediation analysis, which took into account two mediators, supported a relationship between interoceptive sensibility and alexithymia, as well as between alexithymia and emotion dysregulation.

Reactivity ANS is largely connected with the occurrence of alexithymia, particularly with DIF factor, which is supported in numerous reports [66, 67]. A study by Connelly and Denney, [67] indicated that the deficits in emotion dysregulation, as evidenced in those with high levels of the alexithymic trait, appear to manifest as chronically elevated subjective negative affect relative to autonomic activity regardless of the level of environmental demands. Physical symptoms in alexithymia could well be associated with somatosensory amplification [32]. The absence of the capability to name and modulate the affect consciously, by heightening tensions and negative emotions. In our study, we support the hypothesis linking alexithymia with physical problems. These somatoform disorders are the consequence of individual suffering from alexithymia having an inability to appreciate, distinguish, and express affect, which in turn, increases the physiological arousal and those negative subjective states that are not governed by psychological strategies. Friedlander at al., [68] support the speculation from their study that increased physiological arousal and subjective negative emotional experiences in alexithymia sufferers may constitute a cause which leads to poor health. It has been supported in the results of studies by Dubey and Pandey [22] mentioned above.

The results received are consistent with those obtained by other authors who verified the relationship between alexithymia and interoception in people with ASD. Gaigg et al., [58] demonstrated that alexithymia involves a disruption in how physiological arousal modulates the subjective experience of feelings in those with and without a diagnosis of ASD. Their studies support findings that self-report alexithymia questionnaires qualitatively identify extremely alike emotional interoception problems in those with and without a diagnosis of ASD. This is consistent with the observation that self-report levels of alexithymia also predict neural activity in the insula cortex to the same level in those with and without ASD during emotional interoception and empathy tasks [38, 58, 69]. The results of the research cited in the context of the results obtained in our work are critical for another reason. The combination of a self-report alexithymia questionnaire with an experimental study of interoception has shown that self-reported difficulties in identifying and describing one's own emotions are linked to reduced skin conductance responses (SCRs) and independently also with a reduced concordance between subjectively reported and objectively measured levels of arousal in both groups. This supports that at least two distinct processes can contribute to alexithymia–one involving a blunting of emotional experiences and the other involving reduced awareness and cognitive processing of otherwise preserved emotional experiences [41].

The results support the legitimacy of models that consider the perception of physiological arousal (interoception) to play an important role in the subjective experience of feelings [6–8]. Those suffering from alexithymia may either fail to gain awareness of otherwise typical physiological arousal or demonstrate atypical arousal with consequences for the subjective experience of emotions [70]. Ben Shalom et al., [71] concluded that impairments in socio-emotional expression in autism may be related to deficits in perception and/or expression of conscious feelings; physiological emotions may be relatively preserved. The first quantitative integration of results pertaining to the structural neuroanatomical basis of alexithymia made by Xu et al., [72] showed that volumes of the left insula, left amygdala, orbital frontal cortex and striatum were consistently smaller in people with high levels of alexithymia. These areas are important for the perception of emotion and emotional experience. Smaller volumes in these areas might lead to deficiencies in appropriately identifying and expressing emotions. Alexithymia might be associated with the malfunctioning of brain structures, which includes the cingulate and prefrontal cortex regulating and subserving emotional awareness [73] and self-oriented

planning, a mental state often associated with emotion. Hence, it is possible that when mental functions supported by regions in the medial, frontal, and temporal regions fail to work correctly, the orderly linkage from body stimuli to emotions and to symbolic language is disrupted [74] leading to poor symbolization of bodily stimuli and alexithymia.

In addition, such brain activation patterns may be conductive to less frequent use of adaptive, cognitive strategies such as reassessment [75] while emotional dysregulation deficits may in turn lead to more frequent experiencing of negative emotions [75]. The reduced activity of prefrontal areas indicates problems with cognitive top-down processing of emotions in persons showing alexithymic features. It is worth noting that only alexithymia turned out to be an important factor between interoceptive sensibility (ANS measure) and somatoform disorders. Emotional dysregulation only in relation to lack of emotional awareness and lack of emotional clarity. The other aspects were not correlated. It is worth paying attention to this aspect in the preparation of the next research. As research shows, does alexithymia correlate with maladaptive dysregulation strategies and hence is a "more important" predictor in the relationship of somatoform disorder and interoceptive sensibility? Partial verification of this hypothesis may come from the analysis of the results of the longitudinal study carried out by da Silva et al., [76]. Patients diagnosed with a high level of alexithymia during therapy showed a change with regard to one of the emotional variables—lack of emotional awareness, emotion differentiation or emotion regulation. The changes that were observed in the course of therapy in alexithymic patients also related to the had a tendency to focus on physical complaints.

It is worth noting that in the conducted analyses no relationship between body awareness was observed. In the study, we used the BPQ questionnaire, thanks to which it is possible to test body awareness and ANS sensitivity. It is assumed that two styles of attention may alternate. One attention style is associated with hypochondriasis, somatization and anxiety disorders whereas another attention style has been viewed as healthy, adaptive and resilience-enhancing [77]. The ambiguity of the results obtained by us prompts us to conduct further studies on much larger clinical groups and also with the use of other methods to study interoception, e.g. MAIA [78] and monitoring level of anxiety in research.

## Conclusion

People with ASD present higher levels of interoceptive sensibility (autonomic reactivity) as well as alexithymia, and hence higher levels of somatoform disorders in these people, however, the direction of the relationship between the studied factors is similar in both groups. Our study showed that the increased sensibility to interoceptive signals, combined with the incorrect attribution of body signals resulting from alexithymia, increases the level of somatic symptoms. The increased sensibility to interoceptive signals, combined with difficulties in accurately identifying the source of observed symptoms and the inability to differentiate the body's physiological correlates from emotions' physiological correlates, are favorable to the interpretation of symptoms towards a somatic disease. The absence of medical grounds may increase sensibility and result in the higher level of arousal and stress, as well as misinterpretation of somatic condition. Certainly, interventions in order to understand somatic experiences and differentiating emotion correlates and accurate attribution will contribute to the reduction of somatic symptoms.

### Limitations

Our study has some limitations that we would like to highlight. It based exclusively on self-report measures. In addition, it is cross-sectional and correlative. For this reason, it is difficult to clearly outline unequivocal directions or relationships between the investigated factors. We

adopted, in line with the attachment theory, a direction of relationship from interoceptive sensibility through alexithymia to somatic disorders. However, we examined adults and those who, due to somatic problems of unspecified etiology, may show increased alexithymia and interoceptive sensibility. The nature of relationship between emotion understanding and experiencing and becoming sensitive to specific body signals requires longitudinal studies to be carried out in the future. The excessive interoceptive sensibility, combined with alexithymia, may favor the development of somatic dysfunctions. Such is the case with the examined group of individuals with ASD.

## Supporting information

**S1 File.**
(SAV)

## Author Contributions

**Conceptualization:** Elżbieta Zdankiewicz-Ścigała.

**Data curation:** Dawid Ścigała, Wanda Kwaterniak.

**Formal analysis:** Dawid Ścigała.

**Methodology:** Dawid Ścigała.

**Project administration:** Elżbieta Zdankiewicz-Ścigała.

**Resources:** Wanda Kwaterniak.

**Software:** Dawid Ścigała.

**Writing – original draft:** Elżbieta Zdankiewicz-Ścigała, Joanna Sikora, Claudio Longobardi.

**Writing – review & editing:** Elżbieta Zdankiewicz-Ścigała, Claudio Longobardi.

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
