## [Decision Letter · Decision Letter 0]

1 Dec 2020

PONE-D-20-28574

Relationship between Interoception and Somatoform Dissociation in Adults with Autism Spectrum Traits. The Mediating Role of Alexithymia and Emotional Dysregulation

PLOS ONE

Dear Dr. Ścigała,

Thank you for submitting your manuscript to PLOS ONE. After careful consideration, we feel that it has merit but does not fully meet PLOS ONE’s publication criteria as it currently stands. Therefore, we invite you to submit a revised version of the manuscript that addresses the points raised during the review process.

We look forward to receiving your revised manuscript.

Kind regards,

Jane Elizabeth Aspell, PhD

Academic Editor

PLOS ONE

Journal Requirements:

2. Please include your tables as part of your main manuscript and remove the individual files. Please note that supplementary tables should be uploaded as separate "supporting information" files.

5.  We noticed you have some minor occurrence of overlapping text with the following previous publication(s), which needs to be addressed:

- https://pure.royalholloway.ac.uk/portal/files/28092755/150664.full.pdf

- https://www.frontiersin.org/articles/10.3389/fpsyg.2018.01570/full

- https://journals.sagepub.com/doi/10.1177/1362361316667062

- https://www.tandfonline.com/doi/abs/10.1080/09540260500466774

The text that needs to be addressed involves the majority of the Introduction.

In your revision ensure you cite all your sources (including your own works), and quote or rephrase any duplicated text outside the methods section. Further consideration is dependent on these concerns being addressed.

Reviewers' comments:

Reviewer's Responses to Questions

**Comments to the Author**

1. Is the manuscript technically sound, and do the data support the conclusions?

Reviewer #1: Partly

Reviewer #2: Yes

2. Has the statistical analysis been performed appropriately and rigorously? 

Reviewer #1: Yes

Reviewer #2: Yes

3. Have the authors made all data underlying the findings in their manuscript fully available?

Reviewer #1: No

Reviewer #2: No

4. Is the manuscript presented in an intelligible fashion and written in standard English?

Reviewer #1: No

Reviewer #2: Yes

5. Review Comments to the Author

Reviewer #1: There are problems with grammar, clarity and purpose in this manuscript. There is unclarity about the participants' characteristics. Most of the statistical tests are appropriate, and answer the research question. Relevance of cited research is not always clear, while other existing and relevant research is not described/discussed. Conclusion/discussion has a misplaced focus; it appears less based on the authors' current study and more on previous research.

Reviewer #2: Thank you for submitting your interesting work and making contribution to the field. The experimental methods utilised seem reasonable too. Please see some suggestions below.

General comments:

Throughout the manuscript, you have used the term ‘interoception’ as a measure. As you acknowledge too, interoception is a multi-facet domain. You have measured the subjective, self-reported interoceptive ability, which is generally known as ‘interoception sensibility’. Please refer to the literature and use the term consistently throughout the work.

Also, the term dysregulation (when referring to emotional) should be ‘emotional dysregulation’ throughout the manuscript.

Specific comments:

Abstract:

There is duplicate information of participants’ Means and SD (age). Please rectify it.

Introduction:

The introduction section is very long and needs to be concise. Some aspects also needs to be moved around. Suggestions below:

1. Please state aims clearly and much earlier e.g., at the end of Page 3 before you begin the section ‘Interoception and alexithymia in autism’.

2. On page 4, interoception terms and classification (e.g., interoceptive accuracy, interoceptive sensibility and metacognitive awareness; see Garfinkel et al., (2015) or Murphy et al. (2017; interoception and psychopathology)). These need to be defined accurately based on the current knowledge in the field. On the same note, in the beginning of page 5, again, interoceptive accuracy and sensibility are defined in terms of how they measured. Please merge all of these in one place to maintain readers’ flow.

3. The section ‘Dissociation, alexithymia, - regulation of emotions and somatization’ can be made concise too given that some concepts have already been discussed in detail e.g., emotional dysregulation.

Method section:

Please create only one heading of Method starting with ethics details. This should be followed by two sub-sections: Participants and measures.

1. In the participant section, please state age range of both groups separately in text.

2. Measures: I cannot see any reference to how sample size was calculated?

Results:

On page 15, please fix the typos in r values of ‘difficulties in identifying emotions.

Please use interoceptive terms with caution e.g., in the last paragraph. It is important to keep the distinction between bodily awareness and ANS reactivity (also) throughout the manuscript but especially, when discussing and drawing conclusions.

Discussion:

On Page 19, where you discuss alexithymia and interoception in autism, this is an important point. Please refer readers to wider literature e.g., Mull et al. (2018); Shah et al. (2016).

On page 20, literature on brain regions should be used with caution as some of these have very diverse functions; my comment reflects specifically this paragraph, ‘Alexithymia might be associated with the malfunctioning of brain structures, which includes the cingulate and prefrontal cortex regulating and subserving emotional awareness [67] and self-oriented planning, a mental state often associated with emotion. Hence, it is possible that when mental functions supported by regions in the medial, frontal, and temporal regions fail to work correctly, the orderly linkage from body stimuli to emotions and to symbolic language is disrupted [68] leading to poor symbolization of bodily stimuli and alexithymia.

Limitations:

1. When saying requires further validation, it should also include the prospect of more rigorous interoceptive methods to test the hypothesis, as the present study was based on self-reported measures only.

2. The section should be titled ‘limitations and future direction’

3. The sentence, ‘Current observations suggest that alexithymia and dysregulation involves a difficulty in gaining awareness of the body’s state of arousal’ somehow implies causal mechanism. The whole paragraph seems repetitive too. Please consider rewording it.

6. PLOS authors have the option to publish the peer review history of their article (what does this mean?). If published, this will include your full peer review and any attached files.

Reviewer #1: No

Reviewer #2: **Yes: **Farah Hina

---

## [Author Response · Author response to Decision Letter 0]

29 Jan 2021

Thank you very much for the thorough review of our article and considering its for publishing in PLOS ONE. We would like to express our gratitude to you and the reviewers for valuable remarks regarding the manuscript and feedback on how to improve and enhance the content of the manuscript. We really appreciate your remarks and reviewers’ comments, therefore we have modified the text according to the reviewers’ suggestions. Due to the fact that the text has been modified in approximately 70-80 per cent, we have decided to mark modifications in blue colour.

We hope that the changes in the text will meet your expectations, and we will look forward to receiving your response. 

Yours sincerely,

---

## [Decision Letter · Decision Letter 1]

18 Mar 2021

PONE-D-20-28574R1

Relationship between Interoceptive Sensibility and Somatoform Disorder in Adults with Autism Spectrum Traits. The Mediating Role of Alexithymia and Emotional Dysregulation

PLOS ONE

Dear Dr. Ścigała,

Thank you for submitting your manuscript to PLOS ONE. After careful consideration, we feel that it has merit but does not fully meet PLOS ONE’s publication criteria as it currently stands. Therefore, we invite you to submit a revised version of the manuscript that addresses the points raised during the review process.

We look forward to receiving your revised manuscript.

Kind regards,

Jane Elizabeth Aspell, PhD

Academic Editor

PLOS ONE

Journal Requirements:

Reviewers' comments:

Reviewer's Responses to Questions

**Comments to the Author**

1. If the authors have adequately addressed your comments raised in a previous round of review and you feel that this manuscript is now acceptable for publication, you may indicate that here to bypass the “Comments to the Author” section, enter your conflict of interest statement in the “Confidential to Editor” section, and submit your "Accept" recommendation.

Reviewer #1: (No Response)

Reviewer #2: (No Response)

2. Is the manuscript technically sound, and do the data support the conclusions?

Reviewer #1: Partly

Reviewer #2: Partly

3. Has the statistical analysis been performed appropriately and rigorously? 

Reviewer #1: Yes

Reviewer #2: No

4. Have the authors made all data underlying the findings in their manuscript fully available?

Reviewer #1: No

Reviewer #2: Yes

5. Is the manuscript presented in an intelligible fashion and written in standard English?

Reviewer #1: Yes

Reviewer #2: Yes

6. Review Comments to the Author

Reviewer #1: The revised manuscript is much clearer than the first submission. I now have mostly minor recommendations for further improvement.

Reviewer #2: Thank you to authors for resubmitting the article and the amendments. I appreciate all the efforts but have a few concerns, as below.

Abstract:

This statement below needs to be reworded as it seems to be based on speculations (and causation) yet big claims. More to it, previous research investigating somatoform disorders and interoception (Schaefer et al., 2012) has not been mentioned in this study in which there was no evidence of increased interoception in somatoform disorders, as measured by cardiac perception tasks. I acknowledge that the variation in findings when using objective tasks vs self-reported measures. Nonetheless, the findings need to be interpreted with caution (I would suggest the same comment for your conclusion).

“It has been assumed that the interoceptive sensibility is accompanied by a high level of alexithymia and emotion dysregulation, which fosters somatosensory amplification and, as a consequence, results in the intensification of somatoform disorders”

Please avoid the word *proved to be significant*.

Introduction:

This introduction is very long and needs serious amendments (especially, when you consider in comparison to the discussion of findings, which generally needs more elaboration). I admire the authors efforts of providing comprehensive background but this is not a review hence, please make it concise. Right now, it is very hard to follow this introduction. I will expand my comment below:

Your current structure is:

• Interoceptive sensibility and somatoform disorders

• Alexithymia and emotion dysregulation

• Interoception, alexithymia and somatosensory amplification

• Somatoform symptoms (positive and negative) (this and all of the above need to be made concise and in order; you might benefit from subheadings)

• Autism spectrum disorder (ASD); Hypothesis and aims; ASD (All of these need to be re-ordered; hypothesis and rationale should be at the end)

• Interoception and alexithymia in autism; Emotional dysregulation (starting with general interoception, which should be much earlier)

• Neuronal correlates of alexithymia

• Alexithymia and self-reported interoception (this section and the above, neuronal correlates can be made much concise in a few lines)

• Dissociation, alexithymia - regulation of emotions and somatization (contrary to the above, this is an important part of the study; hence the detail makes sense)

• Following this, you again touch upon alexithymia and emotions (deficits in the cognitive processing). This section was already covered in the beginning. The repetition does not make sense or let’s say the order i.e., why would you mention it just before the Method section while it should have been the hypotheses or the study rationale here.

I would suggest you to make subheadings in the following order:

1. Somatoform disorders (including symptoms)

2. Interoceptive sensibility and somatoform disorders (all interoception general details like types or measurement should be covered here too)

3. Aims of the study

4. Autism spectrum disorder and the relevant details

5. Interoception, alexithymia, emotion dysregulation, ASD and dissociation

5.1. Alexithymia and emotion dysregulation

5.2. Interoception and alexithymia (here you can mention neural correlates too, briefly)

5.3. Interoception, alexithymia and emotions in ASD

5.4. Dissociation, alexithymia, emotions and somatization (ASD too if possible)

6. Hypothesis and study rationale

I would re-emphasise that you use sub-headings as this is a lot of information so, try to keep it concise.

Method:

The subheadings with each questionnaire, how its measured and psychometric properties make the section easy to follow.

BPQ: Since you explain autonomic reactivity well, it would be nice to see the same for body perception questionnaire. You might want to add an example or item of these to illustrate especially, for novice readers.

Procedure:

Was the order of completing the questionnaires same or randomised? Please mention

Data analysis:

“In the first place, a series of correlation analyses was performed to assess the relationship between somatoform disorders and other variables”

What other variables? Either name the variables here or write ‘all other’ on which the correlation was carried out.

You have used multiple t-tests, which surely comes with its limitations (that should be mentioned in the limitations section). I would suggest to justify why you chose to do t-tests here.

Results:

Typo: initial analyzes – I assume this is initial analyses. Please correct that. (Same for *more advanced statistical analyzes* just before it says insert figure 2 and 3 here)

“the relationship between interoceptive sensibility (autonomic reactivity) and somatoform disorders was found as regards medium”

How do you define medium? Is it effect size?

“which confirm the direct impact” It might be better to use “support” instead of confirm

Discussion:

When you mention hypothesis, the word “supported” is better than confirmed.

“greater intensity of both interoceptive sensitivity”. I assume you mean interoceptive sensibility. The same comment reflects “increased sensitivity”. Considering interoception literature, these terms can cause confusion. Please use the appropriate terms.

“deeper somatoform symptoms” what do you mean by deeper?

“proved a strong relationship” Please avoid the word prove/d

“These physical illnesses are the consequence of individual suffering from alexithymia having an inability to appreciate, distinguish, and express affect, which in turn, increases the physiological arousal and those negative subjective states that are not governed by psychological strategies”.

Which physical illnesses? Do you mean in general, any named or more so, in terms of somatoform disorders? This seems a claim and needs a reference.

There is too much focus on alexithymia and emotions in autism, which is already quite a common knowledge given the enormous amount of literature available. This diverges the discussion focus from the core area e.g., what are the central elements of this research or in other words, implications? Any possible explanation on the difference found on the autonomic reactivity scale only and not the body perception? Discussion of the findings in relation to the previous research on somatoform disorders and interoception? What next?

Figure 2 (and all other relevant fields related to it): It implies that excessive interoception causes somatoform disorders. What if it’s the other way around? In other words, correlation is not causation. Please review this throughout the manuscript including the results where you mention “direct”

Grammar – please proof read the manuscript. There are several grammar errors, which makes the reading hard.

Overall, throughout the manuscript, there are some strong statements implying causal mechanisms. These should be reworded.

Nonetheless, authors’ efforts are much appreciated.

7. PLOS authors have the option to publish the peer review history of their article (what does this mean?). If published, this will include your full peer review and any attached files.

Reviewer #1: No

Reviewer #2: **Yes: **Farah Hina

---

## [Author Response · Author response to Decision Letter 1]

2 May 2021

We attach once again for your consideration a revised version of our article titled: ‘Relationship between Interoceptive Sensibility and Somatoform Disorder in Adults with Autism Spectrum Traits. The Mediating Role of Alexithymia and Emotional Dysregulation’

We would like to extend our sincere thanks to You and the Reviewers for such positive commentaries on our manuscript and for the helpful proposed revisions. We have carefully considered these revisions and have made changes accordingly and appropriately.

We are extremely grateful for the opportunity you have provided us to revise and re-submit this paper, and believe your reviews and our changes have made the paper stronger.

Yours Faithfully,

---

## [Decision Letter · Decision Letter 2]

17 Jun 2021

PONE-D-20-28574R2

Relationship between Interoceptive Sensibility and Somatoform Disorder in Adults with Autism Spectrum Traits. The Mediating Role of Alexithymia and Emotional Dysregulation

PLOS ONE

Dear Dr. Ścigała,

Thank you for submitting your manuscript to PLOS ONE. One reviewer now recommends acceptance of the manuscript and the other has some further minor recommendations s/he would like addressed (point 6 below). Therefore, we invite you to submit a revised version of the manuscript that addresses the points raised during the review process.

We look forward to receiving your revised manuscript.

Kind regards,

Jane Elizabeth Aspell, PhD

Academic Editor

PLOS ONE

Journal Requirements:

Reviewers' comments:

Reviewer's Responses to Questions

**Comments to the Author**

1. If the authors have adequately addressed your comments raised in a previous round of review and you feel that this manuscript is now acceptable for publication, you may indicate that here to bypass the “Comments to the Author” section, enter your conflict of interest statement in the “Confidential to Editor” section, and submit your "Accept" recommendation.

Reviewer #1: (No Response)

Reviewer #2: All comments have been addressed

2. Is the manuscript technically sound, and do the data support the conclusions?

Reviewer #1: Yes

Reviewer #2: Yes

3. Has the statistical analysis been performed appropriately and rigorously? 

Reviewer #1: Yes

Reviewer #2: Yes

4. Have the authors made all data underlying the findings in their manuscript fully available?

Reviewer #1: Yes

Reviewer #2: Yes

5. Is the manuscript presented in an intelligible fashion and written in standard English?

Reviewer #1: Yes

Reviewer #2: Yes

6. Review Comments to the Author

Reviewer #1: May I congratulate the authors with a much improved and much stronger, very interesting paper. Well done!

Reviewer #2: Well done to authors for improving the manuscript massively. I suggest a few minor changes below:

Introduction:

You have introduced a new model of interoception,

“ IA has been further specified by distinguishing two underlying different, and partially independent, capacities that are included in it: the so-called interoceptive accuracy and interoceptive sensibility [10,11]”.

As the reference remains Garfinkel’s, her model states three dimensions of interoception: interoceptive accuracy, interoceptive sensibility and metacognitive awareness. Hence, please correct your model.

Considering above, it may be helpful to say:

“Interoception refers to the ability to accurately perceive internal bodily processes ….” Instead of “IA is defined as the ability …”

Therefore, please replace “IA” with “interoception” in the context where it applies (especially given that you are measuring interoceptive sensibility in your study).

In the section, “hypothesis and study rationale”, please consider starting with: “It was hypothesized that” paragraph…

Then proceed with rest e.g., justification of the few aspects that you have provided

Typos:

Introduction:

Interoceptive: IA is defined as the ability to accurately perceive interoceptive processes,

Discussion:

i.e., in paragraph 1

In our study, (comma after study)

Thank you for considering the previous comments - congratulations on this interesting piece of research.

7. PLOS authors have the option to publish the peer review history of their article (what does this mean?). If published, this will include your full peer review and any attached files.

Reviewer #1: No

Reviewer #2: **Yes: **Farah Hina

---

## [Author Response · Author response to Decision Letter 2]

18 Jun 2021

Dear Reviewers,

Thank you so much for your time, attention, patience and extensive knowledge. Thank you very much for your all your recommendations. 

Review of revised manuscript.

Reviewer #1: May I congratulate the authors with a much improved and much stronger, very interesting paper. Well done!

Reviewer #2: Well done to authors for improving the manuscript massively. I suggest a few minor changes below

Thank you very much for your acceptance and recognition of our substantive contribution as enough.

Below are comments on minor changes that appeared in Reviewer 2 report

Introduction

“Interoception refers to the ability to accurately perceive internal bodily processes ….” Instead of “IA is defined as the ability …”

Therefore, please replace “IA” with “interoception” in the context where it applies (especially given that you are measuring interoceptive sensibility in your study).

Thank you very much for your correction. We have changed the sentence.

In the section, “hypothesis and study rationale”, please consider starting with: “It was hypothesized that” paragraph

Thank you so much. The clarity of arguments on this change has significantly improved.

Discussion: i.e., in paragraph 1 In our study, (comma after study)

Thank you. The comma has been made.

---

## [Decision Letter · Decision Letter 3]

19 Jul 2021

Relationship between Interoceptive Sensibility and Somatoform Disorder in Adults with Autism Spectrum Traits. The Mediating Role of Alexithymia and Emotional Dysregulation

PONE-D-20-28574R3

Dear Dr. Ścigała,

We’re pleased to inform you that your manuscript has been judged scientifically suitable for publication and will be formally accepted for publication once it meets all outstanding technical requirements.

Kind regards,

Jane Elizabeth Aspell, PhD

Academic Editor

PLOS ONE

Reviewers' comments:

Reviewer's Responses to Questions

**Comments to the Author**

1. If the authors have adequately addressed your comments raised in a previous round of review and you feel that this manuscript is now acceptable for publication, you may indicate that here to bypass the “Comments to the Author” section, enter your conflict of interest statement in the “Confidential to Editor” section, and submit your "Accept" recommendation.

Reviewer #1: All comments have been addressed

Reviewer #2: All comments have been addressed

2. Is the manuscript technically sound, and do the data support the conclusions?

Reviewer #1: Yes

Reviewer #2: Yes

3. Has the statistical analysis been performed appropriately and rigorously? 

Reviewer #1: Yes

Reviewer #2: Yes

4. Have the authors made all data underlying the findings in their manuscript fully available?

Reviewer #1: Yes

Reviewer #2: Yes

5. Is the manuscript presented in an intelligible fashion and written in standard English?

Reviewer #1: Yes

Reviewer #2: Yes

6. Review Comments to the Author

Reviewer #1: (No Response)

Reviewer #2: (No Response)

7. PLOS authors have the option to publish the peer review history of their article (what does this mean?). If published, this will include your full peer review and any attached files.

Reviewer #1: No

Reviewer #2: **Yes: **Farah Hina

---

## [Editor Report · Acceptance letter]

27 Jul 2021

PONE-D-20-28574R3 

Relationship between Interoceptive Sensibility and   Somatoform Disorders in Adults with Autism Spectrum Traits. The Mediating Role of Alexithymia and Emotional Dysregulation 

Dear Dr. Ścigała:

I'm pleased to inform you that your manuscript has been deemed suitable for publication in PLOS ONE. Congratulations! Your manuscript is now with our production department. 

Kind regards, 

on behalf of

Dr. Jane Elizabeth Aspell 

Academic Editor

PLOS ONE